# Shear Wave Velocity to Evaluate the Effect of Botulinum Toxin on Post-Stroke Spasticity of the Lower Limb

**DOI:** 10.3390/toxins15010014

**Published:** 2022-12-26

**Authors:** Yuki Hasegawa, Masachika Niimi, Takatoshi Hara, Yoshihiro Sakurai, Shigeru Soshi, Jun Udaka, Masahiro Abo

**Affiliations:** 1Department of Rehabilitation Medicine, The Jikei University School of Medicine, Tokyo 105-8461, Japan; 2Department of Rehabilitation Medicine, Nihon University School of Medicine, Tokyo 173-8610, Japan; 3Department of Physical Rehabilitation, National Center Hospital, National Center of Neurology and Psychiatry, Tokyo 187-8551, Japan; 4Department of Orthopedic Surgery, The Jikei University School of Medicine, Tokyo 105-8461, Japan; 5Tetsujikai Medical Corporation Kasuga Clinic, Fukuyama 721-0907, Japan

**Keywords:** post-stroke spasticity, botulinum toxin type A, ultrasound elastography, shear wave velocity, gastrocnemius medialis muscle

## Abstract

(1) Background: The evaluation of muscles with spasticity using ultrasound elastography has attracted attention recently, and the shear wave velocity (SWV) technique can measure the mechanical properties of tissues objectively and quantitatively. The purpose of this study was to evaluate the effect of using SWV to assess the effect of Botulinum toxin type A (BoNT-A) treatment in adult patients with post-stroke lower limb spasticity. (2) Methods: We assessed the modified Ashworth Scale, the modified Tardieu Scale, and SWV at rest and after stretching before and at 1 month after BoNT-A treatment in 10 adult participants with post-stroke lower limb spasticity. (3) Results: Significant changes in SWV of the ankle joint in maximum dorsiflexion to the extent possible (SWV stretched) were observed after BoNT-A treatment. SWV stretched was positively correlated with joint range of motion. Participants whose joint range of motion did not improve (i.e., gastrocnemius medialis muscle (GCM) extension distance did not change) had significantly more reductions in SWV stretched after BoNT-A treatment. (4) Conclusions: Our results suggest that the SWV measurements may serve as a quantitative assessment to determine the effect of the BoNT-A treatment in adult stroke patients. SWV measurements to assess GCM spasticity should consider the effects of tension, material properties and activation level of muscles. The challenge is to measure SWV with matching limb positions in patients without contractures.

## 1. Introduction

Post-stroke patients with upper and lower limb hemiparesis may present with spasticity, a clinical sign of upper motor neuron syndrome [1]. Previous reports indicate that spasticity is observed in 19% of patients at 3 months following stroke, and 38% of patients at 12 months following stroke [2,3], and may interfere with daily activities, social participation, and quality of life [4]. Post-stroke lower limb spasticity results in a variety of disadvantages, primarily in gait. Post-stroke lower limb spasticity impairs gait leading to reduced walking speed, increased wheelchair use, and caregiver burden. Adequate joint range of motion (ROM) and muscle strength are essential for walking, but spasticity leads to difficulty in adjusting ROM and muscle tone for walking [5]. In particular, persistent triceps hypertonia causes equinus foot [6], ankle instability during loading phase, and poor toe clearance during the swing phase in gait [7].

Botulinum toxin type A (BoNT-A) treatment, often applied in the cases of spasticity, decreases muscle tone by preventing the release of acetylcholine at the neuromuscular junction [8]. The pharmacological effect of an intramuscular injection of BoNT-A begins 2–4 days following injection, with an expected peak effect at 3 weeks after injection [9]. Several open and placebo-controlled studies have reported the effect of local BoNT-A injections in reducing spasticity, and empathized its safety and ease of clinical use [10,11].

The modified Ashworth Scale (MAS) [12] is utilized often to assess spasticity. In previous reports, MAS has been applied to assess the effect and safety of BoNT-A [10,11,13,14,15]. However, this MAS-based assessment has been demonstrated to be subjective with low inter-rater reliability [16]. Balci et al. suggested that, although assessment with MAS is the most commonly used scale, changes that an evaluator makes in the stretching speed may impact the measurement results because spasticity depends on speed [17].

The modified Tardieu Scale (MTS) [18] is used occasionally as an alternative assessment to MAS. MTS is defined by a constant rate of extension and limb position [19], Some studies have reported MTS to have better retest reliability and inter-rater reliability than MAS in measuring ankle spasticity [20]. However, the MTS is as subjective as the MAS, and the experience of the evaluator may affect reliability [21]. Thus, it is necessary to re-evaluate the evaluation methods used for the proper assessment of spasticity.

In recent years, ultrasonography has been utilized to assess the pathologic muscle changes in patients with lower limb spasticity using the modified Heckmatt scale (MHS) [22]. Higher scores on MHS are associated with a reduced response to BoNT-A treatment [23]. In addition, the evaluation of muscles with spasticity using ultrasound elastography has attracted much attention [24,25]. Within ultrasound elastography, the shear wave velocity (SWV) technique is a noninvasive and objective tool that can directly measure the mechanical properties of tissue [26,27].

Ultrasound elastography is also used to evaluate muscle hardness, mainly in children with cerebral palsy, before and after BoNT-A treatment [28,29]. Previous studies in post-stroke patients have reported a significant reduction SWV in spastic muscles of upper and lower limbs after BoNT-A injection [25,30,31].

Ultrasonographic characteristics (i.e., muscle echo intensity, muscle thickness, posterior pennation angle, Achilles tendon thickness, and sonoelastography) have been used to evaluate effect of BoNT-A treatment with post-stroke spasticity [23,32,33]. Picelli et al. evaluated Achilles tendon stiffness by sonoelastography before and after BoNT-A treatment at approximately 1 month interval, but found no significant changes [33]. Previous studies in post-stroke patients have reported a significant reduction of SWV in Biceps brachii of patients with post-stroke upper limb spasticity after BoNT-A injection [25,31]. However, there is no similar report that evaluates muscles using SWV in lower limb spasticity, thus, it is necessary to consider appropriate measurement positions for the evaluation.

The purpose of this study was to evaluate the use of SWV in assessing BoNT-A treatment in adult patients with post-stroke lower limb spasticity and to determine appropriate measurement positions for the evaluation.

## 2. Results

### 2.1. Study Population Characteristics

Ten patients (7 males) were included in this study. The period since stroke onset was 13.4 ± 3.9 years (mean ± SD) and the age was 62.7 ± 8. years (Table 1). The distribution of age, sex, stroke hemisphere, mechanism, period since stroke onset, MHS, SWV, and date of clinical examination in patients is shown in Table 2.

### 2.2. Changes after BoNT-A Injection and Correlations between SWV and Clinical Examinations

Statistically significant improvements of MAS (*p* = 0.041), R1(*p* = 0.014), and SWV stretched (*p* = 0.005) were observed between before and after BoNT-A injection (Table 3). SWV stretched showed moderate correlations with R2 (*p* = 0.009; ρ = 0.773) before BoNT-A injection after Bonferroni correction (Table 4 and Table 5). 

### 2.3. Comparison between Two Groups

As a result of evaluating the amount of change in SWV by grouping the patients according to the presence or absence of change in ROM (R2), a significant decrease in SWV stretched (*p* = 0.026) was observed in the group with no improvement in R2 (Group 1) (Figure 1).

## 3. Discussion

Although the present study is a small number of cases, there have been no similar studies to examine the effect of BoNT-A treatment on lower limbs spasticity in post-stroke patients by evaluating GCM with SWV. In this study, MAS, R1, and SWV stretched were significantly improved after BoNT-A treatment (Table 3). Shear waves propagate faster through stiffer contracted tissue [26]. Not only stiffness but also active and passive tension influences SWV. Both stiffness and tension increase with increased muscle length and activation [34] and vary depending on the mechanical, material, and structural properties of the muscle [27,35]. The exact relationship between SWV, stiffness, and tension of muscle is not well understood and care must be taken when interpreting SWV [36]. SWV is greatly affected by ankle dorsiflexion angle, and the more the muscle is stretched, the more SWV increases [37,38,39]. Therefore, SWV stretched is more likely to reflect tension than SWV at rest. The significant changes in SWV stretched in the present study may reflect a decrease in skeletal muscle tension that occurs after BoNT-A injection.

The participants in Group 1 showed no improvement in ROM after BoNT-A treatment, suggesting that the joint contractures caused by muscles other than the GCM and soft tissues may have limited the dorsiflexion angle (Figure 1). In Group 2, the increase in passive tension of GCM due to increased dorsiflexion angle may hide the changes of the mechanical properties of the muscle after BoNT-A treatment. Appropriate control of passive tension is considered important for measuring SWV accurately. However, due to the small sample size, this result requires careful interpretation.

Voluntary activation levels, as calculated from electromyography (EMG), affect muscle active tension and stiffness. In healthy young adults, it has been shown that SWV is dependent on activation levels in various muscles such as the gastrocnemius muscles. [40]. Lee et al. quantified the material properties in individual muscles using SWV and % maximum voluntary contraction (MVC), as calculated from electromyography (EMG), of the non-paretic side, the paretic side, and healthy age-matched controls biceps brachii under active conditions. There was a significant difference in the SWV at low levels of activation between non-paretic, paretic, and controls. However, once the activation level increased to 75%, there was no significant difference observed. They hypothesized that any contribution of passive stiffness to total stiffness is negligible once the muscle is activated, and, as such, the increase in passive stiffness of stroke-impaired muscle is inconsequential when measured in an active muscle [36].

Puce et al. investigated EMG activity (spastic dystonia, dynamic stretch reflex, and static stretch reflex) recorded from hypertonic muscles of multiple sclerosis patients. The medial gastrocnemius muscle was not included in the study, but the prevalence of static stretch reflexes ranged from 30% in the soleus muscle to 83% in the radial flexor muscle [41]. Campanella et al. evaluated muscle activation measured by surface EMG and muscle stiffness measured by SWV at approximately 1-month before and after BoNT-A treatment [30]. Following BoNT-A injection, decreases in SWV shortened and stretched conditions and EMG activity indicated that improvement of neuro-mediated reflex hypertonia may have contributed to the decrease in active tension or stiffness. In this study, it is conceivable that BoNT-A treatment improved increase in the neuro-mediated stretch reflex, resulting in a decrease in SWV stretched, which reflects the stretch reflex under the influence of passive tension. However, we did not evaluate muscle activation by EMG and the prevalence rates of various EMG patterns of muscle hypertonia are unknown, estimating whether spastic dystonia and static stretch reflexes affected the SWV results in this study is not possible. Further studies are needed to examine both neuro-mediated hypertonia and intrinsic muscle changes.

SWV at rest is considered to be a method that can minimize the effect of tension. There was a positive correlation between MHS scores and SWV, which was more clear with the muscle in a shortened position [30]. SWV at rest correlated with increased echogenicity and reflects changes in material properties of muscle such as the fibrous-fatty substitution of muscle fibers [35,36]. The fibrous-fatty substitution of muscle fibers has been reported to be a long-term change after BoNT-A treatment [42,43], and we believe that no significant changes could be detected by SWV at rest within 1 month after BoNT-A injection.

In this study, SWV stretched was positively correlated with MTS R2 before BoNT-A treatment (Table 4 and Table 5). SWV may reflect changes in GCM alone. Injections into muscles other than GCM may have affected MAS and MTS results. Some previous studies showed significant correlations between SWV and MAS [28,44,45], while other studies did not [26,39,46]. In situations where long-term spasticity is present, MAS does not necessarily reflect only spastic muscle tone, but must take into account the effects of changes in viscoelastic properties of joint structures and soft tissues [16]. MAS measures a combination of spasticity, contracture, and spastic dystonia. Thus, MAS is not an appropriate assessment of spasticity. Caution should be exercised in the use of MAS and MTS, and the development of other adjunctive assessment tools is important.

This study has several limitations. First, the number of cases in this study is small and the statistical power is insufficient to detect a correlation. Therefore, it is not possible to determine a correlation definitively. Second, this is a single-center study, and patient trends are skewed. In future investigations, we shall consider a classification of participants by time from stroke onset, severity, or function.

Shear wave elastography (SWE) uses an acoustic radiation force impulse, which does not require specific experience of the examiner. SWE is less operator-dependent than strain elastography and represents a reproducible tool for quantifying stiffness [27]. Several studies report good reliability of measurements using SWE in skeletal muscle stiffness [31,47,48,49], however, SWE must be performed properly. In the current study, SWV was measured at the time of its lowest value after maximal dorsiflexion using a standardized protocol among the different evaluators in order to reduce variability.

It has been suggested that the orientation of the transducer should be adjusted so that the SWV measurement is longitudinal to the muscle fibers [50], but it is difficult because the GCM is a pinnate, multifidus muscle. The elasticity of muscle tissue can be evaluated by measurement in the direction along the muscle bundle even in the GCM [51], and the same method was used in this study. Further study is needed regarding the method of measuring SWV.

Since the measurement of ROM with a goniometer is also subject to the subjective judgment of the observer, we shall consider more objective measurements, such as using CPM [52].

## 4. Conclusions

The validity of SWV measurement for the assessment of effect of BoNT-A treatment in patients with post-stroke lower limb spasticity was investigated. The SWV of the stretched GCM showed significant changes, suggesting that SWV measurement may be useful as a quantitative evaluation to determine the effect of BoNT-A treatment on lower limbs spasticity in post-stroke patients. Our results suggest that clinical examination of spasticity such as MAS may lack objectivity in assessing spasticity in the single muscle. The assessment of GCM spasticity by SWV should consider the effects of tension, material properties, and activation level of muscles, and further research is needed. The challenge is to measure SWV with matching limb positions in patients without contractures.

## 5. Materials and Methods

### 5.1. Participants

We recruited post-stroke patients attending the Jikei University Kashiwa Hospital, who received BoNT-A treatment for spasticity of the gastrocnemius medialis muscle (GCM) between August 2021 and March 2022. We evaluated SWV and spasticity results before and at 1 month after BoNT-A injection.

Inclusion criteria for patients with post-stroke spasticity were (i) ≥20 years old; (ii) unilateral cerebral hemispheric lesion by CT or MRI as diagnosed by a radiologist; (iii) residual unilateral motor paralysis (Brunnstrom recovery stage [BRS] ≤ 5); (iv) normal ROM of the ankle joint on the healthy side; (v) no history of cervical radiculopathy or peripheral neuropathy of the lower limb to exclude the effects of neuropathy due to lower motor neuron disease and the associated muscle degeneration; (vi) no history of lower limb trauma or surgery to exclude muscle degeneration due to trauma or surgery; (vii) >1 year since the onset of the disease; (viii) received multiple BoNT-A treatments; and (ix) ≥4 months since the last injection. Exclusion criteria were (i) patients with missing data and (ii) withdrew from the study.

This study was compliant with the Declaration of Helsinki and written informed consent was obtained from all participants. The study was conducted following approval from the institutional ethics committee of the Jikei University School of Medicine (the ethic code [#24-295 (7061)] and expire date: 1 January 2023). This study was registered with the University Hospital Medical Information Network Clinical Trials Registry (UMIN-CTR) (UMIN 000049188).

### 5.2. Ultrasonography

Ultrasonography was conducted by three physiatrists with experience in ultrasonography, and the same conductor performed BoNT-A treatment. Ultrasonography was performed using an Aplio a Verifia system (Canon medical systems, Otawara-shi, Japan), with a 10 Hz linear probe (PLT-1005BT).

In all cases, BoNT-A was injected in the GCM and soleus muscles among the triceps surae muscle. SWE is less robust in deeper muscles as the propagation of the shear waves and, hence, the outcomes depends on the surrounding tissues [27]. Therefore, we chose the GCM for this study.

A B-mode scan was conducted while each patient was in a relaxed supine position with the lower half of the leg out of the bed. All patients were evaluated using the Modified Heckmatt scale (MHS). The MHS is 4-point scale as follows: 1, normal echogenicity in more than 90% of the muscle that is distinct from bone echo; 2, increased muscle echogenicity in 10–50% of tissue, but with distinct bone echo and areas of normal muscle echo; 3, marked increase in muscle echogenicity between 50% and 90% of tissue with reduced distinction of bone echo from muscle; and 4, very strong muscle echogenicity with near complete loss of distinct bone echo from muscle in >90% of tissue [22].

For measurement of SWV at rest (SWV at rest) of the GCM, the participant was prone on a table with knee flexion at 90 o, the lateral side of the lower limb on the side to be measured against the wall, and the ankle free [46]. In order to measure SWV at stretched, the participant was prone on a table, the knee was extended to 0 o, and the ankle joint was slowly and gently extended to the maximum dorsiflexion angle without pain. Each participant was instructed to relax during the examination. We performed measurement in the direction along the muscle bundle [51]. The SWV measurement site was a probe placed in the thickest part of the GCM; the region of interest (ROI) was placed in the muscle belly between the superficial and deep tendinous membrane, excluding skin, bone, blood vessels, or fascia. The ROI size was determined to be 10 mm (H) × 20 mm (W) to remain within the muscle belly according to previously reported procedures [46] (Figure 2). The transducer was placed on top of a layer of acoustic gel without affecting muscle stiffness [46].

### 5.3. BoNT-A Treatment

Onabotulinumtoxin A (BOTOX, Allergan, an AbbVie Company, North Chicago, Illinois, USA) injections were administered to the spastic GCM of all patients under the guidance of ultrasound. The dose of BoNT-A (200–400 units) was adjusted according to the degree of spasticity in each patient. The dose to the GCM, gastrocnemius lateralis, and soleus ranged from 20–70 units (Table 2). The BoNT-A injections were conducted by an experienced physiatrist [45].

### 5.4. Clinical Examination

Three physiatrists performed measurements. They were blinded to the timing of BoNT-A treatment and to the muscle injected. The evaluation before injection and BoNT-A treatment were separated into different days, the evaluator and the persons who performed the injections were separated, and the result of evaluation and treatment were each registered in a highly confidential database, so that each person could not view the data until they were accumulated. Ankle angles were defined as the angle between the lateral midline of the fibula and parallel to the 5th metatarsal (neutral is 90°; plantar flexion <90° to dorsiflexion >90°) [46]. Lower limb spasticity was assessed using MAS and MTS.

The MAS is 6-point scale as follows [12]: 0, no increase in muscle tone; 1, slight increase in muscle tone, manifested by a catch and release or by minimal resistance at the end of the ROM when the affected part is moved in flexion or extension; 1.5, slight increase in muscle tone, manifested by a catch, followed by minimal resistance throughout the remainder (less than half) of the ROM; 2, more marked increase in muscle tone through most of the ROM, but affected part easily moved; 3, considerable increase in muscle tone, passive movement difficult; and 4, affected part rigid in flexion or extension.

The MTS measures spasticity using two parameters [19,21]: quality of muscle reaction (X parameter) and muscle reaction angle (Y parameter). The Y parameters were scored first in V1 (as slow as possible) and then in V3 (as fast as possible). The V1 velocity Y parameter was recorded as the R2 angle, which was defined as the passive range of motion. The V3 velocity Y parameter was recorded as the R1 angle, which was defined as the angle of muscle reaction during fast passive stretch.

The X parameter was scored in the V3 velocity on 5-point scale as follows [19]: 0, no resistance throughout the course of the passive movement; 1, slight resistance throughout the course of the passive movement, with no clear catch at precise angle; 2, clear catch at precise angle, interrupting the passive movement, followed by release; 3, fatigable clonus (<10 s when maintaining pressure) occurring at a precise angle.; and 4, infatigable clonus (>10 s when maintaining pressure) occurring at a precise angle.

The BRS classifies the motor recovery process into 6 stages as follows [52,53]: Stage 1, flaccidity is present and no movements of the limbs can be initiated; Stage 2, the basic limb synergies or some of their components may appear as associated reactions or minimal voluntary movement responses may be present, spasticity begins to develop; Stage 3, the patient gains voluntary control of the movement synergies, although full range of all synergy components does not necessarily develop, spasticity is severe; Stage 4, Some movement combinations that do not follow the synergies are mastered and spasticity begins to decline; Stage 5, more difficult movement combinations are possible as the basic limb synergies lose their dominance over motor acts; and Stage 6, spasticity disappears and individual joint movements become possible.

### 5.5. Statistical Analysis

The collected data were analyzed in SPSS version 28.0 (IBM, Armonk, NY, USA). Each variable of the clinical examinations and SWV were compared between before and after BoNT-A injection with the Wilcoxon signed-rank test.

The data were first assessed for normality using the Shapiro–Wilk test. The age and period since stroke onset, R1, R2, and SWV were distributed normally so they were analyzed using Pearson’s correlation. The score of MHS, MTS, and MAS were analyzed using Spearman’s rank correlation coefficient. The amount of change in SWV before and after BoNT-A injection was analyzed using independent samples *t*-test, dividing the group into two groups, no change in R2 (Group 1) and change in R2 (Group 2). In all calculations, a *p*-value < 0.05 was considered statistically significant. Corrections for multiple comparison were carried out using Bonferroni correction.

### 5.6. Equations

The following equation was used as a guide in the discussion.

The velocity of propagation (Cs) of a shear wave propagating in an isotropic elastic body is defined as
Cs = √(G/ρ) (1)
where G is the shear modulus (modulus of rigidity) and ρ is the density.

In the case of isotropic elastic materials, the following equation holds between Young’s modulus (modulus of longitudinal elasticity) (E) and shear modulus (G), which is generally used as an index of stiffness.
E = 2(1 + v)G(2)
where v is Poisson’s ratio.

Since soft tissue is close to an incompressible medium, Poisson’s ratio is close to 0.5. Therefore, for soft tissue, the relationship between Young’s modulus and shear wave velocity can be approximated by the following equation
E ≅ 3ρ·Cs^2^(3)

In soft tissues, the differences between tissues are small and ρ can be assumed to be approximately 1000 kg/m^3^. Young’s modulus can be estimated from shear wave velocity, and the higher the shear wave velocity, the harder the tissue can be estimated [27,54].

## Figures and Tables

**Figure 1 toxins-15-00014-f001:**
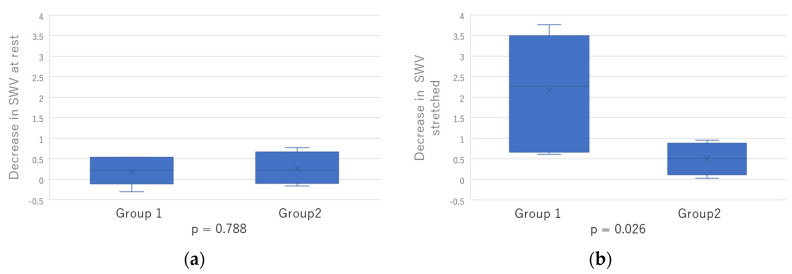
The decrease in SWV before and after BoNT-A injection was analyzed using independent samples *t*-test, dividing the group into two groups: (**a**) Decrease in SWV at rest; (**b**) Decrease in SWV stretched.

**Figure 2 toxins-15-00014-f002:**
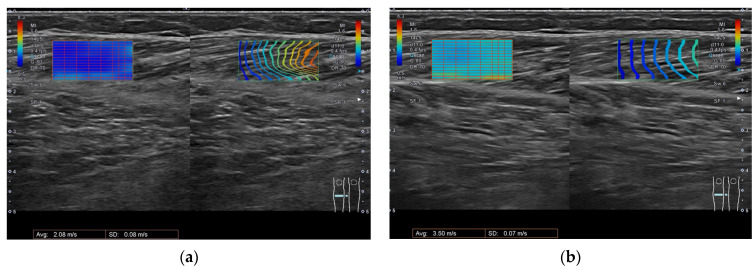
Measurement of the shear wave velocity. The ROI size was determined to be 10 mm (H) × 20 mm in the muscle belly: (**a**) measurement of SWV at rest of the GCM; (**b**) measurement of SWV at extension of the GCM. Left side shows the velocity of shear wave propagation; within the ROI, blue indicates slow propagation and red indicates fast. Right side, the time taken for the shear wave to propagate is indicated by contour lines (arrival time contour lines), which confirm that the shear wave is propagating through the tissue. Blue indicates a small arrival time and red indicates a large arrival time.

**Table 1 toxins-15-00014-t001:** Subject demographics.

Subject Demographics	
Age (years)	
mean (SD)	62.7 (8.7)
Gender (n)	
male/female	7/3
Time since stroke onset (years)	
mean (SD)	13.4 (3.9)
Stroke etiology (n)	
ischemic/hemorrhagic	1/9
ROM changes after treatment: (Group 1) no change; (Group 2) change.	
Group1/2	6/4

**Table 2 toxins-15-00014-t002:** Characteristics of subjects: Abbreviations: period, period since stroke onset (years); dose, BoNT-A dose (units); GCM, gastrocnemius medialis; GCL, gastrocnemius lateralis; Sol soleus; Timing, days to evaluation after BoNT-A injection; MHS, modified Heckmatt scale; MAS, the modified Ashworth Scale before and after BoNT-A injection; MTS X, quality of muscle reaction of the modified Tardieu scale before and after BoNT-A injection; MTS R1, the muscle reaction angle was recorded as fast as possible before and after BoNT-A injection; MTS R2, the muscle reaction angle was recorded as slow as possible before and after BoNT-A injection; SWV at rest, the shear wave velocity at rest before and after BoNT-A injection; SWV stretched, the shear wave velocity stretched before and after BoNT-A injection; F, Female; M, Male; R, right; L, left; H, hemorrhage; I, ischemic.

		Age	Sex	Stroke Hemisphere	Etiology	Period	Dose GCM/GCL/Sol	Timing	MHS	MAS	MTS X	MTS R1	MTS R2	SWV at Rest	SWV Stretched
1	before	56	F	L	H	12.2	50/30/50	30	4	2	2	−15	0	2.49	5.01
	after									2	2	−15	0	2.46	2.82
2	before	60	M	R	H	10.4	30/30/60	28	4	3	2	−30	−20	2.28	6.77
	after									3	2	−30	−20	2.32	3.01
3	before	77	M	L	H	10	40/40/40	29	2	2	3	−10	0	2.6	7.6
	after									1.5	2	−5	0	2.9	5.25
4	before	72	M	L	H	16.3	40/40/40	28	3	2	2	−40	−35	2.23	3.63
	after									2	2	−35	−25	2.16	3.6
5	before	66	F	R	H	21.8	40/40/50	34	3	3	3	−25	−20	2.78	3.86
	after									2	3	−20	−15	2.41	2.9
6	before	62	M	R	I	11.7	40/0/40	28	2	3	2	−40	−35	2.54	2.93
	after									2	2	−30	−20	1.77	2.53
7	before	50	M	R	H	12.1	40/30/40	33	3	2	2	−30	−25	2.17	3.56
	after									2	1	−30	−25	1.64	2.95
8	before	71	F	R	H	17.7	40/0/40	28	3	3	3	−35	−30	1.92	4.14
	after									1.5	1.5	−10	0	2.08	3.5
9	before	59	M	L	H	12.9	20/0/40	29	4	3	2	−30	−10	2.76	6.78
	after									1.5	2	−25	−10	2.36	3.38
10	before	54	M	R	H	9.4	50/50/70	26	3	3	2	−45	−35	2.64	3.21
	after									3	2	−40	−35	2.1	2.52

**Table 3 toxins-15-00014-t003:** Changes after BoNT-A injection changes and correlations between SWV and clinical examinations. Abbreviations: IQR, interquartile range; ^o^, degree; m/s, meters per second; MAS, the modified Ashworth Scale; MTS X, quality of muscle reaction of the modified Tardieu scale; R1, the muscle reaction angle was recorded as fast as possible; R2, the muscle reaction angle was recorded as slow as possible before BoNT-A injection; SWV at rest, the shear wave velocity at rest; SWV stretched, the shear wave velocity stretched. ^a^
*p*-values for continuous variables are form Wilcoxon rank sum tests. * Statistically significant (*p* < 0.05).

	Before BoNT-A Injection	After BoNT-A Injection	*p*-Value ^a^
MAS (Median (IQR))	3 (2–3)	2 (1.5–3)	* 0.041
MTS X (Median (IQR))	2 (2–3)	2 (2–2)	0.102
R1 (^o^) (Median (IQR))	−30 (−40–−22.5)	−27.5 (−31.3–−13.8)	* 0.014
R2 (^o^) (Median (IQR))	−22.5 (−35.0–−7.5)	−17.5 (−25.0–0.0)	0.068
SWV at rest (m/s) (Median (IQR))	2.52 (2.21–2.67)	2.24 (2.00–2.42)	0.093
SWV stretched (m/s) (Median (IQR))	4.00 (3.47–6.77)	2.98 (2.74–3.52)	* 0.005

**Table 4 toxins-15-00014-t004:** Pearson’s correlations between SWV and clinical examinations. Abbreviations: r, Pear-son’s correlation coefficient; *p*, Bonferroni-corrected *p*-value (*p* = 0.0125); Period, period since stroke onset; R1, the muscle reaction angle was recorded as fast as possible; R2, the muscle reaction angle was recorded as slow as possible before BoNT-A injection; SWV at rest, the shear wave velocity at rest; SWV stretched, the shear wave velocity stretched. * Statistically significant, after adjustment for multiple tests using Bonferroni correction.

		Age	Period	MTS R1	MTS R2
SWV at rest	r	−0.106	−0.089	0.227	0.364
before BoNT-A injection	*p*	0.77	0.806	0.528	0.301
SWV at rest	r	0.534	−0.012	0.602	0.579
after BoNT-A injection	*p*	0.111	0.974	0.066	0.079
SWV stretched	r	0.283	−0.325	0.663	* 0.773
before BoNT-A injection	*p*	0.428	0.359	0.036	* 0.009
SWV stretched	r	0.749	−0.069	0.638	0.534
after BoNT-A injection	*p*	0.013	0.85	0.047	0.112

**Table 5 toxins-15-00014-t005:** Spearman Rho Correlation Results between SWV and clinical examination. Abbreviations: ρ, Spearman rho correlation coefficient; *p*, Bonferroni-corrected *p*-value (*p* = 0.017); Period, period since stroke onset; R1, the muscle reaction angle was recorded as fast as possible; R2, the muscle reaction angle was recorded as slow as possible before BoNT-A injection; SWV at rest, the shear wave velocity at rest; SWV stretched, the shear wave velocity stretched; MHS, modified Heckmatt scale; MAS, the modified Ashworth Scale; MTS X, quality of muscle reaction of the modified Tar-dieu scale.

		MHS	MAS	MTS X
SWV at rest	ρ	−0.013	0.355	0.114
before BoNT-A injection	*p*	0.971	0.314	0.754
SWV at rest	ρ	0.243	−0.23	0.634
after BoNT-A injection	*p*	0.498	0.522	0.049
SWV stretched	ρ	0.395	−0.142	0.342
before BoNT-A injection	*p*	0.259	0.695	0.334
SWV stretched	ρ	−0.072	−0.632	−0.201
after BoNT-A injection	*p*	0.843	0.051	0.577

## Data Availability

The data presented in this study are available on request from the corresponding author. The data are not publicly available due to privacy issues.

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
