# Peer review of "Shear Wave Velocity to Evaluate the Effect of Botulinum Toxin on Post-Stroke Spasticity of the Lower Limb"

_toxins, 2022, doi:10.3390/toxins15010014_

Round 1

Reviewer 1 Report

I have major concerns about the scientific soundness of the study design. The sample size was small. The dose injected does not match between Line 237 and Table 1. It appears only medial gastrocnemius received BoNT. However, spasticity in the lateral gastrocnemius and the soleus can also jointly affect the level of spasticity across the plantar flexors and ROM of the ankle. MAS is a relatively subjective scale. Their measurement of ROM changes for MTS appears to be “visual” (Line 173), which is prone to measurement error. It also wasn’t clear to me whether the evaluators were the same physiatrists who performed the injection and the ultrasonography, although it’s mentioned they were blinded; or how they were blinded. The methodological deficits make it hard to interpret the results obtained. Several statistical results were borderline significant. The data did show a significant decrease in “SWV stretched.” However, what that means and how it is clinically correlated were not well manifested. Also, it’s unclear whether the dose difference would affect the results, which becomes another variable. However, it is possible that the manuscript was not well-organized or the above information was not well-described. Otherwise, the authors didn’t explain the metrics/color changes in Figure 2. The tables were not presented in an appropriate manner. There were also some spelling and formatting errors. Therefore, I would suggest rejection. 

Author Response

We thank Reviewer #1 for evaluating our manuscript. The following text describes our response to the comments made by the reviewer.

In this study, BoNT-A was also injected into muscles other than the gastrocnemius medialis muscle (GCM). Only the number of units to the GCM are listed in Table 2 and the Materials and Methods.

For patients with spasticity in the upper and lower limbs, BoNT-A was also administered to muscles other than the GCM. Muscles injected with BoNT-A included the gastrocnemius lateralis and soleus muscles.

The dose of BoNT-A (200–400 units) was adjusted according to the degree of spasticity in each patient. The dose to the GCM ranged from 20-50 units (Table 2).

Clinically, BoNT-A therapy is never conducted with the same dose for all patients. In this study, as well as in the following studies, different doses were used to study the effects on spasticity and function.

We used a goniometer rather than visual assessment to measure joint angles. This is a common technique in MTS evaluations.

We suggest that the lack of correlation between SWV and the clinical evaluation is due to the fact that SWV reflects only the characteristics of the GCM.

As the reviewer pointed out, both MAS and MTS, the two most common assessments used in clinical practice, are fraught with problems. They are both considered to be affected by the lateral gastrocnemius and soleus muscles or soft tissue stiffness. MAS score 3 is also considered to be associated with joint range of motion limitation. In other words, it may include elements of both contracture and spasticity, and we do not believe it is an appropriate assessment of spasticity. However, many BTX trials have been evaluated using these ratings. In other words, the reviewer's point is a fundamental question about past BTX efficacy trials. At the same time, we too would like to address that question. Therefore, we have examined the possibility that elastography in this study may show an auxiliary role as a determination of the effectiveness of BTX.

Regarding the blinding method, the evaluation before injection and BoNT-A treatment were separated into different days, the evaluator and the persons who performed the injections were separated, and the result of evaluation and treatment were each registered in a highly confidential database, so that each person could not view the data until they were accumulated.

Figure 2 (Left side) shows the velocity of shear wave propagation. Within the ROI, blue indicates slow propagation and red indicates fast propagation. (Right side) The time taken for the shear wave to propagate is indicated by contour lines (arrival time contour lines), which confirm that the shear wave is propagating through the tissue. Blue indicates a small arrival time and red indicates a large arrival time.

Reviewer 2 Report

To Authors

Thank you very much for giving me a chance to review this interesting paper to investigate the effect of botulinum toxin on post-stroke spasticity of the lower limb. I assume that the paper is very important for readers in  Toxins, but I have some major comments before publication.

#1: I agree that it is essential to study the effects of botulinum toxin using SWE techniques. I am aware of previous studies because I have researched SWE. Still, the reader needs to review the changes in shear wave velocity or shear elastic modulus in post-stroke spasticity. Therefore, the introduction recommends adding the literature on the change of shear wave velocity or shear elastic modulus in post-stroke spasticity.

#2: There is also a study on the effect of botulinum toxin on post-stroke spasticity using B-mode ultrasound. The novelty of this study will become clearer by describing the difference between this study and those studies.

#3: It is a very good idea to show individual results for Table 2. However, the measurements' timing is unclear, so please correct the table to clarify the contents.

#4: This study aimed to examine botulinum toxin's effect, not to examine the relationship between shear wave velocity and each index before and after the intervention. This is not necessary for the present study since the results have been described in other previous studies. Alternatively, a correlation analysis of the change before and after the intervention could be considered.

#5: In the present study, the ROM changes were used to divide the patients into groups, but due to the small sample size, the validity of this study is questionable.

#6:Please add the reliability of the shear wave velocity measurement.

#7: In the discussion, I would like to see an argument as to why botulinum toxin reduced the shear wave velocity. In addition, I would like to see a paragraph describing the clinical application of botulinum toxin and how this change can be clinically interpreted.

I am looking forward to seeing the revised paper soon.

Author Response

We thank Reviewer #2 for evaluating our manuscript. The following text describes our response to the comments made by the reviewer.

#1: Content regarding evaluation using shear wave elastography in post-stroke spasticity has been added to Line 99-111.

#2: Content regarding ultrasonographic characteristics in post-stroke spasticity has been added to Line 99-111. Although the present study is a small number of cases, there have been no similar studies to examine the effect of BoNT-A treatment on lower limbs spasticity in post-stroke patients by evaluating GCM with SWV , suggesting an meaningful outcome. Following the biceps brachii study [Gao et al.2019], we also consider it a novelty to evaluate spastic muscles by extending the muscle.

Gao, J.; Rubin, J. M.; Chen, J.; O'Dell, M. Ultrasound Elastography to Assess Botulinum Toxin A Treatment for Post-stroke Spasticity: A Feasibility Study. Ultrasound Med Biol. 2019, 45(5), 1094-1102; DOI: 10.1016/j.ultrasmedbio.2018.10.034.

#3: We added in Table 2 how many days after BoNT-A treatment we evaluated for each patient.

#4: There has been no report of a method to measure GCM using SWV as an evaluation of BoNT-A treatment of lower leg spasticity, and it is necessary to examine the appropriate measurement limb position at rest or stretched.
We have also performed a correlation analysis of the changes before and after the intervention as you suggested, but it was not statistically significant (SWV at rest ρ = 0.347 p = 0.295, SWV stretched ρ = -0.458 p = 0.156).
We noticed that SWV stretched was significantly affected by tension. We believe that it is important to establish a testing method first, and the discussion focused on how to interpret SWV stretched and at rest, which is why we are performing the analysis as in this study.
The entire discussion was revised from LINE 249 to LINE 317 because it was difficult to understand.

#5: We thought that this analysis was visually clear in showing that appropriate control of muscle tension was considered important for measuring SWV accurately.
However, as the reviewer pointed out, we have added a note to LINE 263-264 that ‘due to the small sample size, this result requires careful interpretation.’

#6:The following was added to Discussion Line 331-464.
Shear wave elastography (SWE) uses an acoustic radiation force impulse, which does not require specific experience of the examiner. SWE is less operator-dependent than strain elastography and represents a reproducible tool for quantifying stiffness [27]. Several studies report good reliability of measurements using SWE in skeletal muscle stiffness [31,47, 48, 49], however, SWE must be performed properly. In the current study, SWV was measured at the time of its lowest value after maximal dorsi-flexion using a standardized protocol among the different evaluators in order to reduce variability.
It has been suggested that the orientation of the transducer should be adjusted so that the SWV measurement is longitudinal to the muscle fibers [50], but it is difficult because the GCM is a pinnate, multifidus muscle. The elasticity of muscle tissue can be evaluated by measurement in the direction along the muscle bundle even in the GCM [51], and the same method was used in this study. Further study is needed regarding the method of measuring SWV.

#7: We assumed that SWV would be primarily affected by passive and active tension, and material properties of muscles.
The discussion on passive tension has been added to discussion LINE 248-264.
The discussion on active tension has been added to discussion LINE 265-307.
The discussion on SWV at rest and material properties of muscle has been added to discussion LINE 308-315.

Reviewer 3 Report

The manuscript discusses the data obtained from 10 patients with post-stroke ankle spasticity assessed with shear wave elastography and clinical scales before and after botulinum toxin A treatment. The results confirm that shear wave velocity on the stretched gastrocnemius medialis muscle significantly changes after treatment and provides further information about the joint angle and patients' age.
The study is well conducted although the main results are not particularly innovative.
Several major and minor issues are here reported:

Abstract: MTS abbreviation is used without a prior definition.

Abstract: specify that the evaluated gastrocnemius muscle is the medial, as said in the methods.

Introduction line 28: spasticity is not a symptom, but rather a clinical sign.

Table 2 is very difficult to read.

Tables 4 and 5: why prefer "Sig." instead of the usual "p" for statistical significance?

Results lines 112-116: it is not clear to me why SWV stretched, which decreases very significantly after BoNT treatment (table 3), has been analysed again considering the amount of pre-post treatment change and separating subjects with and without ankle ROM variation, to show a marginally significant difference of SWV only in those with unchanged ROM.

Methods line 197 "unilateral cerebral hemispheric lesion by CT or MRI as diagnosed by a radiologist": why is it specified "by a radiologist" (or rather neuroradiologist)? The lesion should be related to the previous stroke and in a motor area, of course not all unilateral brain lesions cause spasticity.

Methods lines 199-200 "no history of cervical radiculopathy or peripheral neuropathy of the upper limb": Why?

Methods line 200 "no history of upper limb trauma or surgery": why?

Methods line 201 "received multiple BoNT-A treatments": why multiple BoNT-A treatments are required?

Methods line 203 "the leg against the wall": which leg? Contralateral? The lateral part against the wall?

Methods line 225: strictly speaking, the SWV measured while keeping ankle plantar flexor muscles stretched does not reflect spasticity (velocity-dependent, occurring during passive movement, etc) but rather the static phase of the stretch reflex, which is recordable by EMG only in some of the patients with the upper motor neuron syndrome (static stretch reflex is reported in the soleus of about 30% of patients with multiple sclerosis, see Puce et al, 2021, DOI 10.1016/j.cnp.2021.05.002 ). The lack of concomitant EMG recording does not allow the correct characterization of muscle hypertonia in the evaluated patients, so data interpretation should be more carefully discussed. It is useful and appropriate to discuss also a recently published manuscript assessing SWV in patients with spastic dystonia following stroke (Campanella et al, 2022, DOI 10.3389/fneur.2022.980746 )

Methods line 241 "Three physiatrists performed measurements": muscle stretching for SWV assessment was also performed by different evaluators? If so, this is a source of variability due to different stretching modalities and strength. This affects not only parameters derived from the clinical evaluation but also SWV.

Methods lines 253-258: MTS description should be clarified, particularly regarding the meaning of "dynamic muscle length".

Discussion lines 120-121 "The decrease in SWV stretched represents a change in the mechanical properties of the GCM": how this sentence is justified by the data? What do the Authors mean by "mechanical properties"? Intrinsic muscle stiffness? How intrinsic changes can occur after BoNT injection?

Author Response

We thank Reviewer #3 for evaluating our manuscript. The following text describes our response to the comments made by the reviewer.

#1: Abstract: specify that the evaluated gastrocnemius muscle is the medial, as said in the methods.
#1: The relevant section of Line 19, 26 and 478 has been changed.

#2: Introduction line 28: spasticity is not a symptom, but rather a clinical sign.
#2: The relevant section of Line 30 has been changed.

#3: Table 2 is very difficult to read.
#3: Table 2 was modified to make the before and after changes easier to understand, and the timing of measurements and dose of gastrocnemius lateralis and soleus were also added.

#4: Tables 4 and 5: why prefer “Sig.” instead of the usual “p” for statistical significance?
#4: For Tables 4 and 5, we changed the notation from Sig. to p.

#5: Results lines 112-116: it is not clear to me why SWV stretched, which decreases very significantly after BoNT treatment (table 3), has been analysed again considering the amount of pre-post treatment change and separating subjects with and without ankle ROM variation, to show a marginally significant difference of SWV only in those with unchanged ROM.
#5: We assumed that SWV would be primarily affected by passive and active tension, and material properties of muscles.
The discussion on passive tension has been added to discussion LINE 248-264.
The discussion on active tension has been added to discussion LINE 265-307.
The discussion on SWV at rest and material properties of muscle has been added to discussion LINE 308-315.
We thought that this analysis was visually clear in showing that appropriate control of muscle tension was considered important for measuring SWV accurately, and the increase in tension of GCM due to increased dorsiflexion angle may hide the changes of the mechanical properties of the muscle after BoNT-A treatment. We has been added to discussion LINE 258-264. However, we have added a note to LINE 263-264 that ‘due to the small sample size, this result requires careful interpretation.’ 

#6: Methods line 197 “unilateral cerebral hemispheric lesion by CT or MRI as diagnosed by a radiologist”: why is it specified “by a radiologist” (or rather neuroradiologist)? The lesion should be related to the previous stroke and in a motor area, of course not all unilateral brain lesions cause spasticity.
#6: Almost all brain imaging studies in our hospital are diagnosed by a neuroradiologist to confirm that it is a stroke and to rule out multiple sclerosis or other diseases other than stroke presenting with spasticity.

#7: Methods lines 199-200 “no history of cervical radiculopathy or peripheral neuropathy of the upper limb”: Why?
#7: Corrected text to “lower limb” instead of “upper limb”. 
To exclude the effects of neuropathy due to lower motor neuron disease and the associated muscle degeneration. 

#8: Methods line 200 “no history of upper limb trauma or surgery”: why?
#8: Corrected text to “lower limb” instead of “upper limb”.
The modified Heckmatt scale evaluation of muscle brightness is also to exclude muscle degeneration due to trauma or surgery. 

#9: Methods line 201 “received multiple BoNT-A treatments”: why multiple BoNT-A treatments are required? 
#9: Many of our hospital patients have received their first BoNT-A treatment at other hospitals, in order to match conditions between patients.

#10: Methods line 203 “the leg against the wall”: which leg? Contralateral? The lateral part against the wall?
#10: The lateral side of the lower leg on the side to be measured. Added on LINE 559.

#11: Methods line 225: strictly speaking, the SWV measured while keeping ankle plantar flexor muscles stretched does not reflect spasticity (velocity-dependent, occurring during passive movement, etc) but rather the static phase of the stretch reflex, which is recordable by EMG only in some of the patients with the upper motor neuron syndrome (static stretch reflex is reported in the soleus of about 30% of patients with multiple sclerosis, see Puce et al, 2021, DOI 10.1016/j.cnp.2021.05.002 ). The lack of concomitant EMG recording does not allow the correct characterization of muscle hypertonia in the evaluated patients, so data interpretation should be more carefully discussed. It is useful and appropriate to discuss also a recently published manuscript assessing SWV in patients with spastic dystonia following stroke (Campanella et al, 2022, DOI 10.3389/fneur.2022.980746 )
#11: It would be most important that the significant changes in SWV stretched in this present study may reflect a decrease in skeletal muscle tension that occurs after BoNT-A injection (LINE 248-306).
However, we believe the reviewer's point is correct. Because we did not evaluate muscle activation by EMG and the prevalence rates of various EMG patterns of muscle hypertonia are unknown, it cannot be estimated whether spastic dystonia and static stretch reflexes affected the SWV results in this study. We have cited the study presented and added Limitation LINE 294-306.

#12: Methods line 241 “Three physiatrists performed measurements”: muscle stretching for SWV assessment was also performed by different evaluators? If so, this is a source of variability due to different stretching modalities and strength. This affects not only parameters derived from the clinical evaluation but also SWV.
#12: In this study, SWV was measured using a standardized protocol even among different evaluators, and the SWV was measured at the time of its lowest value after maximal dorsiflexion, in order to reduce variability. We have added a note to limitation LINE 330-336.

#13: Discussion lines 120-121 “The decrease in SWV stretched represents a change in the mechanical properties of the GCM”: how this sentence is justified by the data? What do the Authors mean by “mechanical properties”? Intrinsic muscle stiffness? How intrinsic changes can occur after BoNT injection?
#13: We consider "mechanical properties" to be stiffness and tension. 
We consider that BoNT-A treatment improved increase in the neuro-mediated stretch reflex, resulting in a decrease in SWV stretched, which reflects the stretch reflex under the influence of passive tension. 
This can be justified by the result that SWV at rest, which is not affected by tension, remained unchanged, while SWV stretched decreased significantly. We have added a note to DISCUSSION LINE 249-310

Reviewer 4 Report

(1)Please provide the full names of the abbreviations when they first appear in the abstract and text. 

E.g. MAS, MTS in Abstract.

(2)Please check the manuscript carefully to remove the typos, improve the language and format. E.g.

The fonts in Para. 1, Section 1 are not uniform.

It is better if Table 2 has black frame/

(3)It is better to move Section 5 to Section 2. Results should follow Methodology.

(4)The authors should use more equations to specify the Methodology, and use more mathematical technologies in analysis.

Author Response

We thank Reviewer #4 for evaluating our manuscript. The following text describes our response to the comments made by the reviewer.

(1)Please provide the full names of the abbreviations when they first appear in the abstract and text. 
E.g. MAS, MTS in Abstract.
I have revised the points pointed out by the reviewer.

(2)Please check the manuscript carefully to remove the typos, improve the language and format. E.g.
The fonts in Para. 1, Section 1 are not uniform.
It is better if Table 2 has black frame/

I have revised the entire document by correcting the points pointed out by the reviewer.

(3)It is better to move Section 5 to Section 2. Results should follow Methodology.

This order could not be changed because it follows Toxins formats.

(4)The authors should use more equations to specify the Methodology, and use more mathematical technologies in analysis.

Corrections for multiple comparison were carried out using Bonferroni correction. The entire DISCUSSION has been revised because significant correlations are almost gone.

The following equation was used as a guide in the discussion.
The velocity of propagation (Cs) of a shear wave propagating in an isotropic elas-tic body is defined as 
Cs=√(G/ρ) (1)
Where G is the shear modulus (modulus of rigidity) and ρ is the density.
In the case of isotropic elastic materials, the following equation holds between Young's modulus (modulus of longitudinal elasticity) (E) and shear modulus (G), which is generally used as an index of stiffness.
E = 2(1+v)G (2) 
Where v is Poisson's ratio.
Since soft tissue is close to an incompressible medium, Poisson's ratio is close to 0.5. Therefore, for soft tissue, the relationship between Young's modulus and shear wave velocity can be approximated by the following equation
 E ≅ 3ρï½¥Cs2 (3) 
In soft tissues, the differences between tissues are small and ρ can be assumed to be approximately 1000 kg/m3. Young's modulus can be estimated from shear wave veloc-ity, and the higher the shear wave velocity, the harder the tissue can be estimated [27, 54].

Round 2

Reviewer 2 Report

Thank you for submitting the revised manuscript.

I have no further concerns on my part. I think this is a very interesting study and I look forward to your future work.

Congrats!!

Reviewer 3 Report

The revised manuscript significantly improved and is suitable for publication.

Reviewer 4 Report

Accept.